# Agent-based simulation for reconstructing social structure by observing collective movements with special reference to single-file movement

Hiroki Koda[1]*, Zin Arai[2], Ikki Matsuda[2,3,4,5]

**1** Primate Research Institute, Kyoto University, Inuyama, Aichi, Japan, **2** Chubu University Academy of Emerging Sciences, Kasugai, Aichi, Japan, **3** Wildlife Research Center of Kyoto University, Kyoto, Japan, **4** Japan Monkey Centre, Inuyama, Japan, **5** Institute for Tropical Biology and Conservation, Universiti Malaysia Sabah, Kota Kinabalu, Malaysia

* koda.hiroki.7a@kyoto-u.ac.jp

**Data Availability Statement:** The code used in the study are available from: https://github.com/hkoda/mix_gauss_caravan_sna. https://doi.org/10.5281/zenodo.3876449.

## Abstract

Understanding social organization is fundamental for the analysis of animal societies. In this study, animal single-file movement data—serialized order movements generated by simple bottom-up rules of collective movements—are informative and effective observations for the reconstruction of animal social structures using agent-based models. For simulation, artificial 2-dimensional spatial distributions were prepared with the simple assumption of clustered structures of a group. Animals in the group are either independent or dependent agents. Independent agents distribute spatially independently each one another, while dependent agents distribute depending on the distribution of independent agents. Artificial agent spatial distributions aim to represent clustered structures of agent locations—a coupling of "core" or "keystone" subjects and "subordinate" or "follower" subjects. Collective movements were simulated following two simple rules, 1) initiators of the movement are randomly chosen, and 2) the next moving agent is always the nearest neighbor of the last moving agents, generating "single-file movement" data. Finally, social networks were visualized, and clustered structures reconstructed using a recent major social network analysis (SNA) algorithm, the Louvain algorithm, for rapid unfolding of communities in large networks. Simulations revealed possible reconstruction of clustered social structures using relatively minor observations of single-file movement, suggesting possible application of single-file movement observations for SNA use in field investigations of wild animals.

## Introduction

Identifying latent structures underlying animal social organization is fundamental in animal ecology [1], as well as in the social sciences [2]. In order to infer the biological mechanisms underlying the emergence of complex animal societies, researchers have traditionally recorded animal social data in the multiple dimensions of animal societies; for example, spatial distances

**Funding:** This study was mainly funded by the Japan Science and Technology Agency, Core Research for Evolutional Science and Technology 17941861 (#JPMJCR17A4) and partly by MEXT Grant-in-aid for Scientific Research on Innovative Areas #4903 (Evolinguistics), 17H06380. The funders had no role in study design, data collection and analysis, decision to publish, or preparation of the manuscript.

**Competing interests:** NO authors have competing interests.

among group members, social relationships (male–female or mother–offspring relationships), and movements based on group decision making.

The recent update of computational models for social network analysis (SNA) enables quantitative evaluation of complex processes of social networks based using big data. The basic framework of SNA envisions multiple nodes as animals, viruses, people, or any other agents within the network and at its edges as a dyadic relation mathematically defined between the two nodes (e.g., [3–5]). The SNA aims to statistically link network structures with bottom-up rules defining dyadic relations, e.g., affiliations, friendships, proximities, co-occurrences, follower/followee, signal sender/receiver, or communications [6]. SNA provides a modern powerful interface for visualizing complex biological networks.

One major biological interest for SNA is statistical approximations of modular, clustered or hierarchical structures embedded in animal social groups [1, 7]. Doubtless, kinship is a basic concept in the theory of the socioecology (for a classical kickoff, e.g., [8]). A mother invests effort in her offspring, mother and offspring generally maintain proximity, and degrees of the social bonding are reflected by relatedness [9, 10]. Based on the cumulative effects of such relatedness, social structures are determined by additional social relations, such as male-female sexual interactions and male-male competition [11, 12]. Consequently, a wide variety of social roles in the animal group are recognized, and multi-layered clustered structures of the group emerge [1, 13–15]. Some metrics commonly implemented in the modern SNAs provide statistical indices for modularity, centrality or cohesiveness (e.g., [16]), and have been applied to actual observational data of animals.

For the practical applications of SNA to observational data, dyadic social interactive behaviors or distance-based proximities have been commonly used in ecological analyses of animal societies [3, 17]. Allogrooming is a typical example of dyadic social interaction that is widely observed in social mammals. Allogrooming is an altruistic behavior and an appropriate behavioral metric for evaluating affiliative interactions (e.g., [18–20]). Likewise, agonistic interactions, where dominant animals show aggression toward subordinate animals, are also interpretable and countable behaviors that can be used to characterize dominance hierarchies (e.g., [21–23]). Generally, in the case of dyadic social interactions, researchers calculate the occurrence frequencies of dyadic events using an observational rule; for example, counting the grooming occurrences and recording the identity of the groomer and groomee in a specific observational session. These data obtained for social interactions can be used to construct matrices of social relationship strength by scoring or weighting the obtained data of dyadic relationships [17]. However, despite the interpretable advantages of social relationships, there are certain disadvantages associated with such data. Behavioral-based metrics such as grooming or fighting are generally unsuitable for the automation of data acquisition or evaluation. Currently, at least, researchers must record animal behaviors via direct observation or video recording for data analysis. Additional disadvantages are the "specializations" of social behaviors; for example, grooming might not always be observed in insects, or we may have little knowledge regarding the agonistic interactions among some animals, thereby limiting the analysis. In contrast, distance-based data lack contextual information and are not perfectly linked to social relationships [24]. In the case of distance-based proximities, inter-individual distances are recorded using a regular cycle of sampling rates (e.g., recording the distance at 5-min intervals), and proximities are evaluated. Subsequently, a matrix of social relationships is generated in a similar manner, scoring or weighting the proximity. Given that close proximity is assumed to indicate an affiliative relationship, the thresholds of the specific distance defining the social "association" can be used to generate binary data (i.e., association or isolation), and an adjacency matrix is generated from a summation of the binary data [17]. These data are generally accepted to indirectly reflect social

affiliations [25]. Affiliated individuals may maintain proximity as typically seen in mother-offspring pairs. The recent advanced tracking methods using high-precision GPS (e.g., [26, 27]) or Bluetooth beacons (e.g., [28]) have strongly influenced location-based proximity metrics due to dramatic expansion of data for interactions among "trackable" animals. Real-time distance-based proximities among all animal agents empirically reveal core mechanisms underlying collective movement emergencies. Simultaneous tracking of multiple animals using wearable tags is technologically possible. However, perfect tracking of suitable numbers of animals in groups/units for analyses of collective movement are still limited to laboratory-based animals (e.g., [28, 29]) or capturable wild animals [30–32]. Thus, a key analytical step for SNA is selection of appropriate data, which researchers record massively, easily, reliably and precisely.

The main objectives of this study were (1) to propose a novel approach for obtaining observable data that has not been systematically applied in the estimation of animal social organization, and (2) to provide methodological validation for the proposed approach. The first objective is based on practicality, particularly with respect to those studying wild animals. Dyadic social interactive behaviors or distance-based proximities have been a golden standard used to infer animal social organizations, as mentioned above; however, those studying animal behavior often experience difficulty in obtaining large amounts of high-quality data for such dyadic social interactions or spatial proximity in wild animals. For example, it is extremely difficult to monitor complete behavioral activities, even if wearable sensors are available. Similarly, in most cases, experimental manipulations, such as artificial control of specific individual movements, are not possible for wild animals (for a discussion of the methodological issues, see [17]). Accordingly, researchers invariably search for alternative promising types of data, together with technological updates. For the generalized application of novel types of data, empirical evidence that can be used to validate the approaches is necessary, which could partly be achieved using numerical simulations.

In this study, another kind of testable data for the SNA is proposed, that is, "animal single-file movement" generated by bottom-up rules in the theory of collective movements. This topic is currently of great interest in animal ecology (for a recent review, e.g., [33]). Typically in fish schools, bird flocks, insect swarms, or ungulate herds, large numbers of individual agents show highly-coordinated collective movements [34, 35]. Theoretically, collective movements do not rely on decision-making of specific individuals, e.g., a "leader", but emerge in a bottom-up manner such as iterated interactions between animals, either directly (e.g. via visual cues) or indirectly (e.g. via trail formation).

Animal single-file movement, defined as serially ordered patterns of animals that are indirectly determined by degrees of inter-agent relations, is a promising target for data collection for linking SNA linking with recent discussions of collective movement by social animals. First, single-file movements are observable in many free-ranging animal species. Classically, social animals moving in single file provided an excellent opportunity to count group members inhabiting dense forests with poor visibility, and further to infer social structures. Extensive historical fieldwork reveals that such events provide information on latent key factors underlying animal social organizations, e.g., dominance hierarchy [36, 37], kin relationships [38], and physiological status, such as breeding cycle [39, 40]. Additionally, due to the recent human imposed threats such as roads, animals are often forced into single-file formation to cross risky or dangerous places. Such an aspect of socio-spatial organization under highly risky situations has also been analyzed in meerkat [41] and elephant [42]. In fact, recent studies focusing on single-file movement and social system structure have examined several animal taxa, e.g., wolves, *Canis lupus*: [36]; zebra, *Equus burchellii*: [39], buffalo, *Syncems cuffer*: [40], chimpanzee, *Pan troglodytes*: [43], mandrill, *Mandrillus sphinx*: [44]

and spider monkey, *Ateles geoffroyi*: [45]. Second, animal single-file movement can be recorded without capture and release for attachment of location tags, simply by video-recording (e.g., camera-trappings). This advantage for studying collective movement avoids use of GPS collars that are a burden for free-ranging animals, and most cannot be used for threatened and endangered species [46]. Also, single-file movement is naturally observed even in zebra heading to water holes [39] or in wolves through snow [47]. Anywhere such single-file movements are often observable recently developed camera-trapping techniques [48] would enable collection of a valuable data set. Further, where animal movement is controllable in semi-free range conditions such as narrow gateways to feeders and water, another opportunity exists. Such single-file movement has been recorded for sheep, *Ovis aries*: [37]. Third, serial order patterns governed by collective movements allow behavioral ecologists to examine individual-based observation essential for documentation and measurement of animal behavior. Fourth, the methodological advantages listed above compensate for the disadvantages of other metrics commonly used in the previous studies, such as dyadic social interactive behaviors or distance-based proximities.

This study aimed to determine if movement order in single-file movements are informative and effective observations for the reconstruction of animal social structures using an agent-based computational simulation model, e.g., [49, 50]. An initial artificial 2-dimensional spatial distribution was developed using the simple assumption of clustered group structures. Animal agents in a group are composed of independent and dependent agents. Independent agents distribute spatially independently each other, while dependent agents distribute depending movements of independent agents. The artificial spatial distributions aim to represent clustered structures of agent locations—a coupling of "core" or "keystone" subjects and "subordinate" or "follower" subjects. Collective movements were governed by two simple rules, 1) initiators of the movement are randomly chosen, and 2) the next moving agent is always the nearest neighbor of the last moving agents. These bottom-up rules allowed simulation of "single-file movement" data. Finally, we visualized social networks and reconstructed clustered structures using a recent major SNA algorithm, the Louvain algorithm, for rapid unfolding of communities in large networks [51]. The current idea—that artificial agents are initially generated, collective movements are produced, and collective movements are used to evaluate social networks—is similar to a previous computer experiment [52]. The current simulation is more generalized. Parameter assumptions are minimum and used mixed normal distributions. Parameter conditions were sought where single-file movement data work well for reconstruction of the latent clustered structures and further discuss possibilities for practical applications in field investigations of wild animal societies.

## Materials and methods

The simulations described below can be used to examine whether individual-by-individual serially ordered movements, i.e., "animal single-file movements," provide sufficient information to reconstruct the clustered organization that is typically believed to exist in animal social groups, such as those of primates. Our simulations adopt agent-based models and involve four main steps: (1) animal agents are distributed in two-dimensional space with latent cluster organizations; (2) these generated agents are serially aligned following a simple collective movement rule (generating the "single-file movements"); (3) the social network structure is computed using an association index defined by serial orders of animal single-file movement; and (4) the cluster organization of the generated social network is evaluated. Fig 1 shows a schematic representation of the simulation process. The details of each step are described below.

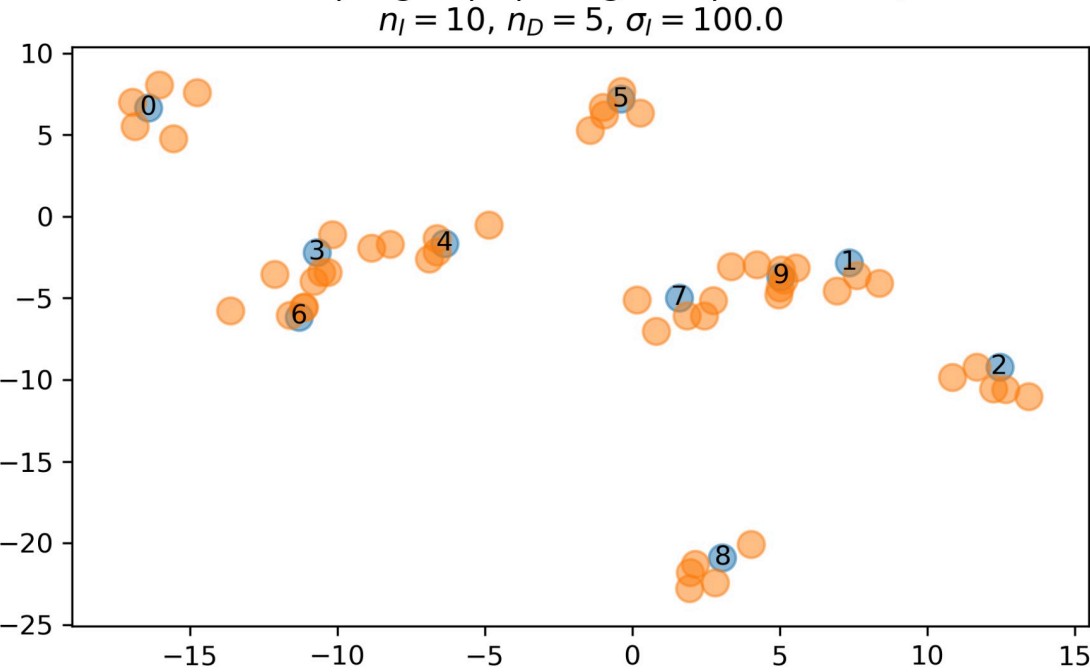

**Fig 1. Schematic illustration of the simulation process.** In this example, the parameters were set as follows: $n_I = 5$, $n_D = 5$, $\sigma_I =$ 10, $\sigma_D = 1$, $n_{\exp} = 10$. The simulation started from the spatial distributions of agents generated from one latent state of social group organization, determined by parameters of the mixed Gaussian processes, $n_I$, $n_D$, $\sigma_I$, $\sigma_D$ (the two top layers). The "single-file movement" data sets were then generated using the process described in the section on ordered alignment of agents. The vertical chains of the 30 circles represent single-file movement (the number of each circle is the agent id). Adjacency matrices $G_k$ for $k \in$ $\{1, 2, \ldots 10\}$ were generated from orders in the single-file movement data set, and were convolved to a single adjacency matrix $\mathbf{G}$, which was passed to the social network analysis. Finally, a social network graph was produced, with clustering of the local community (bottom graph). This flow is a single simulation process, which is run 1000 times.

## Distributing animals (1$^{st}$ step)

We initially assume that the animal group of interest consists of $n_I$ subgroups. Each of these subgroups has a unique *independent* animal agent (*independenter*), who moves independently of any other agent, and $n_D$ *dependent* animal agents (*depender*), the spatial locations of which are invariably dependent on the location of the independenter of the subgroup. For simplicity, the number $n_D$ of dependers in subgroups is assumed to be the same for all subgroups. Accordingly, the total number of animal agents is $N = n_I(n_D + 1)$. We refer to the independenter of the $i$-th subgroup as the $i$-th independenter.

The locations of independenters are independently generated from the bivariate Gaussian distribution: the location of the $i$-th independenter in two-dimensional space is denoted by $\boldsymbol{x}_i \in \mathbb{R}^2$ and is distributed as

$$\boldsymbol{x}_i \sim \mathcal{N}(\mathbf{0}, \Sigma_I) \tag{1}$$

for $i \in \{1, \cdots n_I\}$. Here $\Sigma_I := \begin{pmatrix} \sigma_I^2 & 0 \\ 0 & \sigma_I^2 \end{pmatrix}$ is the covariance matrix of the distribution ($\sigma_I$ is the standard deviation of the independenter). The mean of the distribution is set to $\mathbf{0}$, the origin of the plane.

Similarly, the locations of dependers are generated from the bivariate Gaussian distribution, but with centers dependent on the locations of independenters. The location of the $j$-th

### Estimated social networks with parameters, $n_I = 10$, $n_D = 5$, $\sigma_I = 100.0$, $n_{exp} = 10$ Score = 1.0

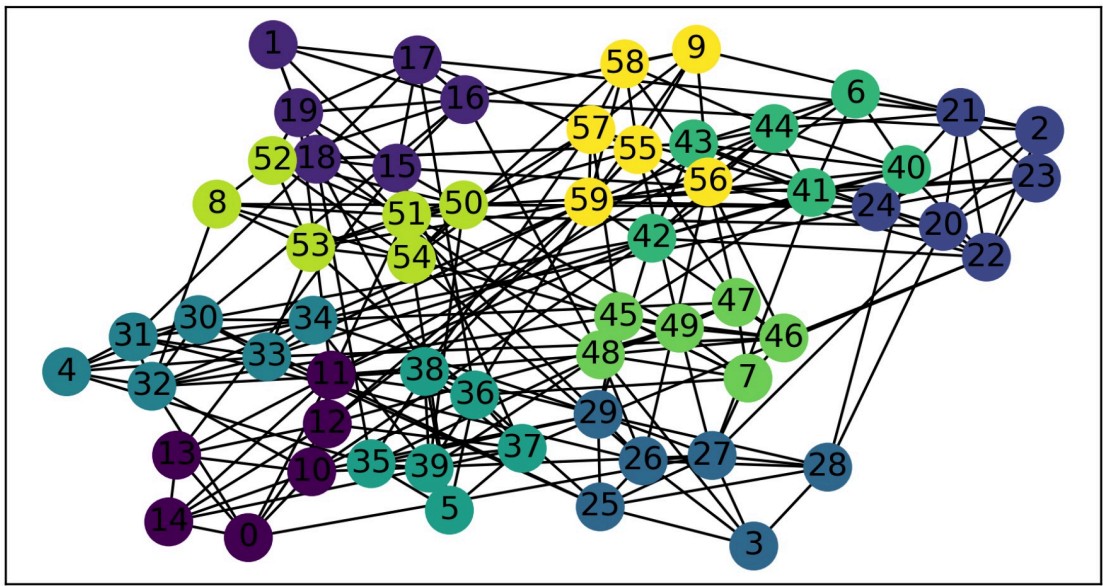

**Fig 2. Example of the agent spatial distribution generated with model parameters, $n_I = 10$, $n_D = 5$, $\sigma_I = 100.0$.**

depender for the $i$-th subgroup is denoted by $\boldsymbol{x}_k$ where $k = n_I + (i \times j)$ and are distributed as

$$\boldsymbol{x}_k = \boldsymbol{x}_{n_I+(i\times j)} \sim \mathcal{N}(\boldsymbol{x}_i, \Sigma_D), \tag{2}$$

for $i \in \{1, \cdots n_I\}$ and $j \in \{1, \cdots n_D\}$. Here $\Sigma_D := \begin{pmatrix} \sigma_D^2 & 0 \\ 0 & \sigma_D^2 \end{pmatrix}$ is the covariance matrix of the distribution ($\sigma_D$ is the standard deviation of the depender). The mean parameters are set to each independenter's locations, $\boldsymbol{x}_i$. For simplicity, we always assume that $\sigma_D = 1$.

Consequently, the spatial distributions of $N$ animal agents in the plane is dependent on three parameters, $n_I$, $n_D$, and $\sigma_I$. Fig 2 shows the sample spatial distribution with the following parameter values: $n_I = 10$, $n_D = 5$, and $\sigma_I = 100.0$.

## Serialization ($2^{nd}$ step)

Subsequently, "animal single-file movement," the ordered alignment set of animal agents, is generated assuming a simple rule of collective movements defined by the mutual distance between agents. The process is iterative: (1) the agent who first moves (*initiator*) is randomly selected; (2) in each iteration step, the nearest agent to the last moved agent is selected to move next. In this iterative process, the order of the single-file movement is defined as the order of movement.

More specifically, the initiator $\boldsymbol{y}_1$ is randomly selected from a multi-set of $N$ agents

$$A_1 = \{\boldsymbol{x}_1, \ldots, \boldsymbol{x}_N\}. \tag{3}$$

The next moving agent $\boldsymbol{y}_2$ is selected by

$$\boldsymbol{y}_2 = \arg\min_{\boldsymbol{x}_k \in A_2} \|\boldsymbol{y}_1 - \boldsymbol{x}_k\| \quad \text{where} \quad A_2 = A_1 \setminus \{\boldsymbol{y}_1\}. \tag{4}$$

Similarly, the $i$-th moving agent is determined by

$$\boldsymbol{y}_i = \arg\min_{\boldsymbol{x}_k \in A_i} \|\boldsymbol{y}_{i-1} - \boldsymbol{x}_k\| \quad \text{where} \quad A_i = A_{i-1} \setminus \{\boldsymbol{y}_{i-1}\}. \tag{5}$$

Here, $\|\cdot\|$ represents the Euclidean distance on the plane. The process of agent selection is repeated $N-1$ times. Consequently, a totally ordered set is generated:

$$C := \{\boldsymbol{y}_1, \ldots, \boldsymbol{y}_N\}, \tag{6}$$

which we refer to as "animal single-file movement." Note that the process of generating $C$ depends on the three parameters, $n_I$, $n_D$, and $\sigma_I$.

We repeat the process of generating the distribution of animals (1st step) and constructing the animal single-file movement (2nd step) $n_{\exp}$ times. Here, we implicitly refer to $n_{\exp}$, the number of observation opportunities for single-file movement in the field. In the following step, we therefore assume $n_{\exp}$ takes an experimentally feasible value, say, $n_{\exp} = 10$, or 30.

Finally, we denote the set of $n_{\exp}$ single-file movements by $\boldsymbol{C} := \{C_1, \ldots, C_{n_{\exp}}\}$.

## Generating a social network ($3^{rd}$ step)

In this step, we construct a weighted graph representing the social network structure of animals using the information of $\boldsymbol{C}$.

The graph is defined by an $N \times N$ adjacency matrix $\boldsymbol{G}$ constructed from $\boldsymbol{C}$ as follows: First, given an animal single-file movement $C$, we define an adjacency matrix $G$ as follows:

$$G = \begin{pmatrix} a_{11} & \cdots & a_{1i} & \cdots & a_{1N} \\ \vdots & \ddots & & & \vdots \\ a_{i1} & & a_{ii} & & a_{iN} \\ \vdots & & & \ddots & \vdots \\ a_{N1} & \cdots & a_{Ni} & \cdots & a_{NN} \end{pmatrix} \tag{7}$$

where $a_{ij} = a_{ji}$ is 1 if the agents $\boldsymbol{x}_i$ and $\boldsymbol{x}_j$ are adjacent in $C$, and 0 otherwise (by definition, all diagonal elements are 0). The adjacency matrix $\boldsymbol{G}$ for $\boldsymbol{C}$ is defined as

$$\boldsymbol{G} := \sum_{k=1}^{n_{\exp}} G_k \tag{8}$$

where $G_k$ is the adjacency matrix for $C_k$, and is the $k$-th single-file movement in $\boldsymbol{C}$.

For the implementation of this step, we use a recently widely used framework for social network analysis, `NetworkX` (https://networkx.github.io/documentation/stable/index.html) in `python3` based on the `from_numpy_matrix()` method (https://networkx.github.io/documentation/stable/reference/generated/networkx.convert_matrix.from_numpy_matrix.html). We note that the *parallel_edges = False* option is set, and the entries in $\boldsymbol{G}$ are interpreted as the weight of a single edge joining the vertices.

## Clustering ($4^{th}$ step)

Finally, we run a graph clustering algorithm to examine whether the subgroup structure of animal agents can be reconstructed from $\boldsymbol{G}$. For this purpose, we apply the Louvain algorithm,

which is one of the greedy optimization algorithms of modularity. The algorithm is implemented in the Python module `community` (or `python-louvain` on pypy) as its method `community.best_partition` (see https://python-louvain.readthedocs.io/en/latest/api.html). We denote the number of clusters in the result of the Louvain algorithm by $n_{eval}$. Fig 3 represents an example of a social network graph with clustering obtained using our procedure.

## Evaluation of the method

Here, we describe our procedure for estimating the accuracy of the clustering we obtained. For this purpose, we simply count the number of clusters obtained, $n_{eval}$, and compare this with the "correct answer" and the number of subgroups $n_I$. Namely, we consider the ratio $r = n_{eval}/n_I$ and refer to this as the score of the cluster estimations. When $r = 1$, our procedure successfully recovers the information of the number of clusters, whereas $r > 1$ or $r < 1$ implies an overestimation or underestimation, respectively. For each combination of parameters $n_I \in \{1, 2, \cdots 20\}$, $n_D \in \{1, 2, \cdots 20\}$, $n_{exp} \in \{10, 30\}$, and $\sigma_I \in \{1.0, 2.0, 5.0, 10.0, 100.0\}$, we run the entire procedure 1000 times and compute the average $\bar{r}$ of the score $r$. The result are discussed in Section Experimental Results.

## Remarks on the method

As described in section Experimental Results, the proposed method works reasonably well for most of the selected parameter values. In this subsection, we attempt to explain why this is the case by showing that the modularity of the "true" clustering is likely to be higher than that of the clustering obtained by attaching two clusters of the true clustering together. Given that the Louvain algorithm optimizes the modularity through a process of inductive amalgamation of clusters, if the true clustering has this property, the algorithm can be expected to generate the true clustering, which implies that the score $r$ will be 1.

We begin by recalling the definition of modularity. Consider a weighted graph $G$ with $n$ nodes defined by a symmetric $n \times n$ weighted adjacency matrix $G = (a_{ij})$ and a clustering of $G$ into mutually disjointed sets of nodes $C_1, C_2, \ldots, C_{n_I}$ where $n_I$ is the number of clusters. The modularity of this clustering is defined as

$$Q = \frac{1}{2M} \sum_{i,j=1}^{n} \left( a_{ij} - \frac{k_i k_j}{2M} \right) \delta(C(i), C(j)) \tag{9}$$

where $k_i$ and $k_j$ are the sum of the weights of edges from nodes $i$ and $j$, respectively; $M$ is the sum of the weights of all edges in the graph; $C(i)$ and $C(j)$ are the clusters to which nodes $i$ and $j$ belong, respectively, and $\delta$ is the Kronecker delta function. Thus, $\delta(C(i), C(j))$ is 1 if $i$ and $j$ belong to the same cluster but otherwise 0. Note that $2M = \sum_{i,j} a_{ij}$. For $\boldsymbol{G}$ construction in the 3rd step of our method, if an animal single-file movement has $N - 1$ edges, $M = (N - 1) n_{exp}$ holds.

If we assume that $\{C_1, C_2, \ldots, C_{n_I}\}$ be the true clustering, that is, the clustering exactly reflects the original subgroup structure of animals, and denote its modularity by $Q$, and then consider a clustering $\{C_1, \ldots, C_p \cup C_q, \ldots, C_{n_I}\}$ obtained from the true clustering by attaching $C_p$ and $C_q$ together and denote its modularity by $Q'$, a direct calculation shows that

$$Q' - Q = \frac{m}{M} - \frac{K_p K_q}{2M^2} \tag{10}$$

where $m$ is the sum of the weights for all edges from $C_p$ to $C_q$; $K_p$ and $K_q$ are the sum of the weights for all edges from $C_p$ and $C_q$, respectively. We want to show that this quantity is

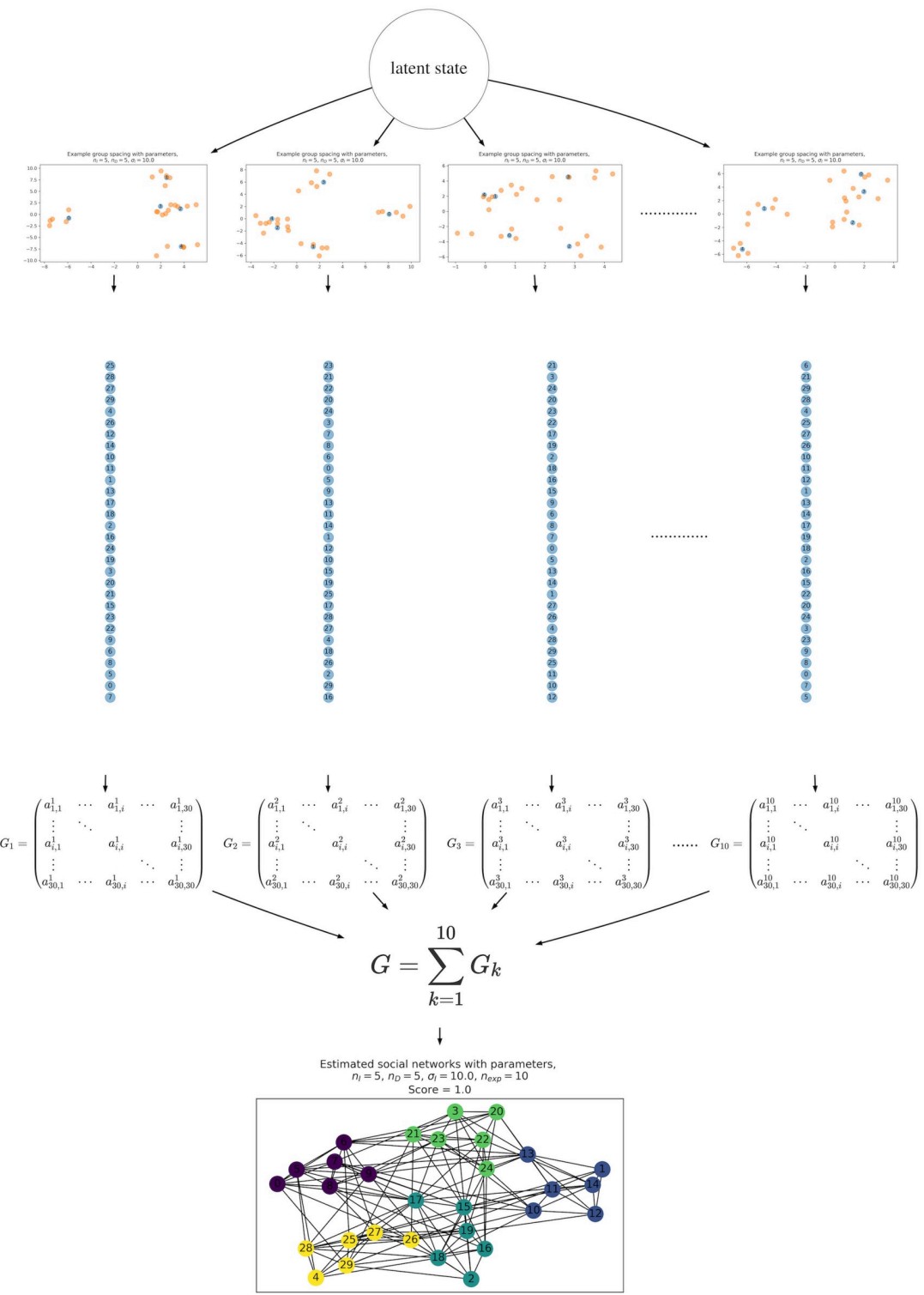

**Fig 3. Example of a social network graph clustered using the Louvain algorithm.** This graph was generated from the spacing patterns of animal agents shown in Fig 2.

negative; that is,

$$m < \frac{K_p K_q}{2M}. \tag{11}$$

For simplicity, we assume that in each single-file movement, no mixture of subgroups occurs; that is, the first $n_D + 1$ agents in the movement belong to the same subgroup, and the next $n_D + 1$ belongs to another one, and so on. This is more or less equivalent to assuming that $\sigma_I$ is very large, and therefore each each subgroup is completely separated. In this case, (11) can be reasoned as follows: A single-file movement assigns $n_D$ edges to the first and last clusters and $n_D + 1$ edges to others. Therefore, $K_p, K_q \leq n_D n_{\exp}$ follows. Hence, a sufficient condition for (11) to hold is as follows:

$$m < \frac{(n_D n_{\exp})^2}{2M} = \frac{n_D^2 n_{\exp}}{2(n_I(n_D + 1) - 1)} = \frac{n_D}{2} \cdot \frac{n_D}{(n_D + 1) - 1/n_I} \cdot \frac{n_{\exp}}{n_I}. \tag{12}$$

In contrast, the number of edges between agents that belong to different subgroups is $n_I - 1$ for each single-file movement. Therefore, the sum of the weights for edges in $G$ connecting different subgroups is $(n_I - 1) n_{\exp}$. Thus, the expected value of $m$ is

$$\frac{(n_I - 1) n_{\exp}}{n_I(n_I - 1)} = \frac{n_{\exp}}{n_I} \tag{13}$$

provided that all the ordering of subgroups in single-file movements is equally probable. The right-hand side of (12) can now be understood as the expected value of $m$ multiplied by

$$\frac{n_D}{2} \cdot \frac{n_D}{(n_D + 1) - 1/n_I}, \tag{14}$$

which is approximately $n_D/2$ in our parameter settings. This implies that, when $n_{\exp}$ is sufficiently large, (12) is very likely to hold for all values of $p, q$.

## Experimental results

Fig 4 is the contour plots of the calculated average score $\bar{r}$ for some selected sets of parameter values. This suggests that the average score increases as parameters $\sigma_I$ and $n_{\exp}$ increases, just as expected. It is also observed that the score decreases as $n_I$, the numbers of the independenter, increases. In contrast, it increases as $n_D$, the numbers of the depender, increases. In summary, scores will be better with large $\sigma_I$, $n_{\exp}$, $n_D$ and small $n_I$ (expect for the case of $n_I = 1$). In the cases of $n_{\exp} = 30$ and $\sigma_I \geq 5$, scores were often perfect, i.e., $\bar{r} \approx 1$. Thus, cluster size estimations were often successful, if animal single-file movements were observed 30 times.

## Discussion

### Result interpretations

Simulations with agent-based models suggest that recent implementation of a community clustering algorithm in the SNA could well estimate latent community structures generated by mixed Gaussian distributions under several parameter conditions. First, SD ratio parameters, $\frac{\sigma_I}{\sigma_D}$, likely influence performance for identification of clustering structures. The greater these ratios are, the more precise the estimations of cluster size obtained. This result is reasonable since the contrast of the modular structure becomes clear (see Fig 2 as an example of the $\sigma_I = 100$). Likewise, the more iterations, the more accurate scores of clusters size became. This result is also not surprising. An important result is the effect of numbers of *dependers* for the cluster estimation.

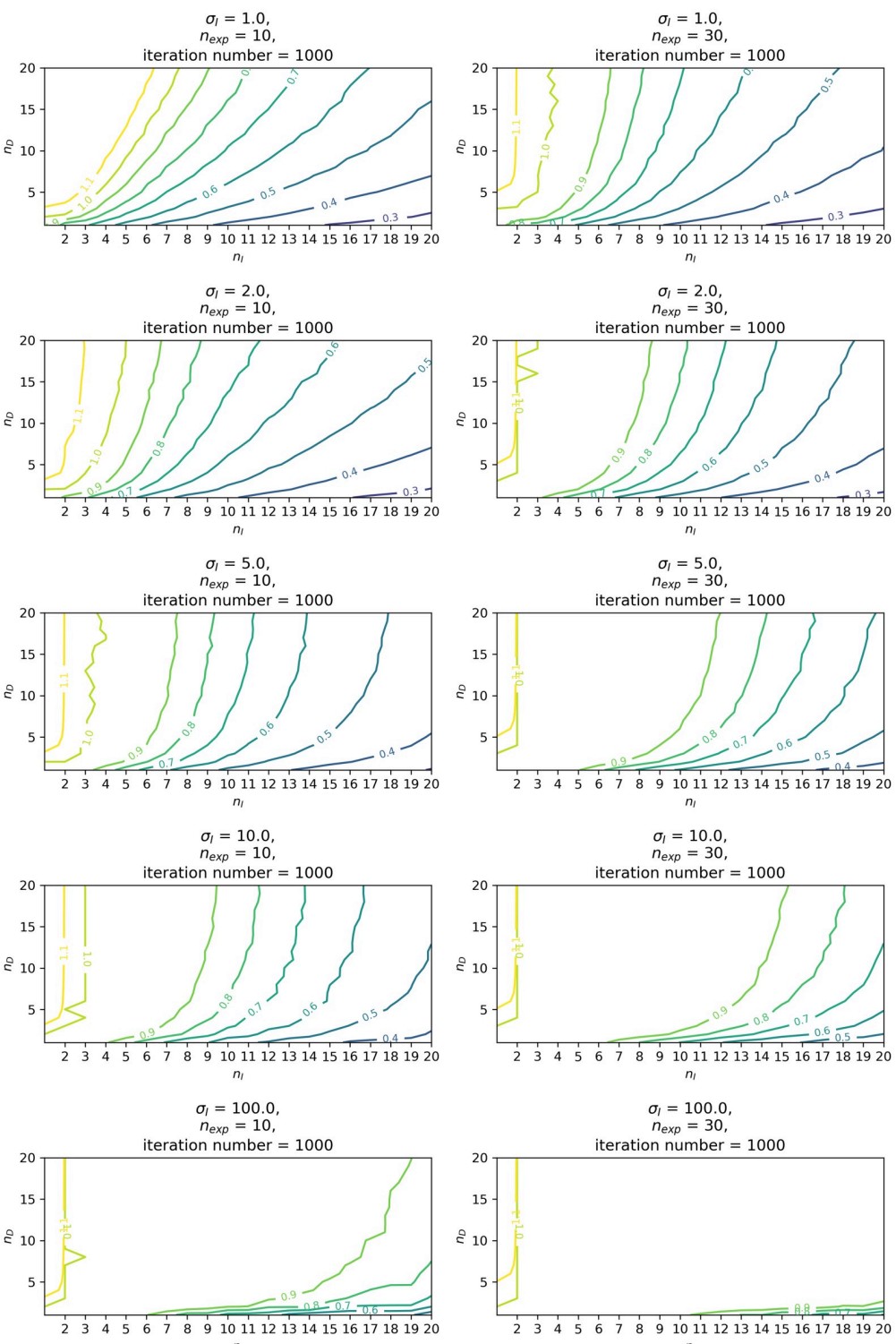

**Fig 4. Contour plots for average performance scores for cluster estimation, $\bar{r}$ in 1000-steps simulations.** Areas near the score of 1.0 represents areas of parameter combinations where perfect estimation scores returned.

Results consistently indicate that greater numbers of *dependers* improved estimation performance, at least within the range of parameter settings used in study simulations. Too many *dependers* would likely lead to overestimations of the cluster sizes, but such confusion occurred only in the case of $\sigma_I = 1$, $n_{exp} = 10$, and did not occur using most other parameter combinations.

An additional lesson from the simulations is an implication that appropriate numbers of the *independenters* exist for better performance in cluster estimation. At least for parameter settings of $1 \leq n_I \leq 30$, a single-digit number, except for 1 and 2, likely shows better performance. In summary, results suggested that SNA applications of the Louvain algorithm to single-file movement data work well even when using only 30 iterations in simulations ($n_{exp} = 30$), when 1) the *independenters* range between around 3 and 10, 2) *dependenters* count is as large as reasonable, and 3) SD ratios of *independenter* per *dependenter* are high.

## Practical applications: Primate societies as a possible candidate

Looking back to observations of actual animal societies in reality, animals would fit with conditions appropriate for fine reconstruction using the present method? A promising candidate, for example, are primate societies.

Primates are middle to large mammals mainly inhabiting tropical forests and sometimes adapting to dry savanna, temperate forests or cool-temperate climates [53]. Primates show a large diversity of social systems [54], from simple monogamous to complex multi-level societies that presumably emerged based on the ecological constrains, such as competition for food and threats of predation, as discussed in successful ecological models, e.g., [55–57]. Their societies are the most investigated animal society from both theoretical and empirical perspectives, and the "social roles" or "social structure" have been commonly recognized under the ecological constrains, e.g., an "alpha" subject, the most dominant social rank, or the kin-based clusters, such as the female affiliative bonding [57, 58]. Thus, principles of primate social formation rely on kin-based relationships and due to ecological constrains, group size is typically limited to about 100 individuals, but mostly groups range from 2.5-50 individuals [59]. Group size and structure, which are often consist of a few matrilineal clustered families and a few independent males, seemingly match with the successful parameter conditions indicated in the simulations.

Another advantage of primates is methodological characteristics of primate behavioral ecology. Primates are typically long-lived animals with slow life histories, and long-term studies that have been ongoing continuously for approximately a decade or more, are relatively common [60, 61]. Traditionally, primatologists have made individual long-term observations. Consequently, all group members can be perfectly identified in both field and captive studies. Observations at road or river crossings—a suitable event for single-file movement—have been reported in field observations for several primate species, e.g., chimpanzee [62], mandrill [44] and proboscis monkeys [63]. Such events (single-file movement) would potentially happen frequently in the field, though these observations may be used only for counting group members. Relevant findings of simulations are that the SNA works well for only 30 iterations of the single-file observation, and such observations may well exist for well-studied primates. Given advantages in primate groups, applications of the present SNA to their single-file movement data can be empirically tested soon. The quest to find the social structures of the primates is a fundamental issue, studied by many pioneer primatologists. Thus, the proposed approach provides a novel and empirical tool to analyze primate society, especially for their socio-spatial organization.

## Next for the methodological generalizations

We must acknowledge that our proposed simulation approach has certain limitations. Firstly, we examined only a very limited number of parameter assumptions regarding animal social

organization. For example, the number of *dependenter* per *independenter* is always constant, and the "clustering parameters," represented by the SD ratio of $\frac{\sigma_I}{\sigma_D}$, were also assumed to be constant for the *dependenters* and *independenters*. Moreover, the simulated single-file movements were observed without any misidentification of the passing agents. However, it is often difficult to reproduce the behavior of wild animals based on the strong and simple assumptions characterizing the present simulations. For methodological generalization, we must further evaluate the simulated groups of the variable sets of the parameters, or a simulated case of the single-file movement in which the identities of some agents are masked.

Ideally, the method should be validated using realistic animal data. As mentioned above, camera trapping would be a promising interface to automatically collect data on single-file movements. For primate researchers engaged in longitudinal field projects, animal identification is possible by observing video clips obtained using camera traps.

Finally, we also acknowledge possible integration with other social network data in primate societies. The output from the single-file movement-SNA is just one dimension of inference for social structures. For a deep understanding of social organizations, investigation of animal society with multidimensional views and data is needed. For example, genetic test of relatedness is a direct evaluation of network patterns. Behavioral observations, e.g., traditional quantitative grooming counts of the groomer-groomee or vocal exchanges of callers and responders, are also direct metrics that reflect social networks. Gut bacterial flora of groups of individuals was recently suggested as a possible candidate to elucidate the network structures of animal society. Thus, access to many kinds of elemental data is available for predictions concerning social networks. Together with comparisons among network structures reconstructed by different metrics, an ultimate understanding of fundamental mechanisms underlying the animal society will eventually emergence.

## Acknowledgments

We appreciate the profound comments and suggestions from Ichiro Tsuda, Aru Toyoda, and Takashi Morita. This study was performed under the Cooperative Research Program at KUPRI (2018-C-27, 2019-B-27).

## Author Contributions

**Conceptualization:** Hiroki Koda, Ikki Matsuda.

**Data curation:** Hiroki Koda.

**Formal analysis:** Hiroki Koda, Zin Arai.

**Funding acquisition:** Hiroki Koda, Ikki Matsuda.

**Investigation:** Hiroki Koda.

**Methodology:** Hiroki Koda, Zin Arai.

**Project administration:** Hiroki Koda, Ikki Matsuda.

**Resources:** Hiroki Koda.

**Validation:** Hiroki Koda.

**Visualization:** Hiroki Koda.

**Writing – original draft:** Hiroki Koda, Zin Arai, Ikki Matsuda.

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
