## [Decision Letter · Decision Letter 0]

12 Oct 2020

PONE-D-20-20675

Agent-based simulation for reconstructing social structure by observing collective movements with special reference to single-file movement

PLOS ONE

Dear Dr. Koda,

Thank you for submitting your manuscript to PLOS ONE. After careful consideration, we feel that it has merit but does not fully meet PLOS ONE’s publication criteria as it currently stands. Therefore, we invite you to submit a revised version of the manuscript that addresses the points raised during the review process.

We look forward to receiving your revised manuscript.

Kind regards,

Ilya Safro, Ph.D.

Academic Editor

PLOS ONE

Journal Requirements:

Reviewers' comments:

Reviewer's Responses to Questions

**Comments to the Author**

1. Is the manuscript technically sound, and do the data support the conclusions?

Reviewer #1: Partly

Reviewer #2: Yes

2. Has the statistical analysis been performed appropriately and rigorously? 

Reviewer #1: Yes

Reviewer #2: Yes

3. Have the authors made all data underlying the findings in their manuscript fully available?

Reviewer #1: Yes

Reviewer #2: Yes

4. Is the manuscript presented in an intelligible fashion and written in standard English?

Reviewer #1: Yes

Reviewer #2: Yes

5. Review Comments to the Author

Reviewer #1: Very interesting work. The paper is well written, and all the four steps of your systems are clear.

Counting the number of clusters and compare it with the the ground truth seems too naive, however, I do think this a good starting point, given the experimental nature of this work.

A stronger experimental section would help. For example, it is always useful to see the scalability of your system.

Reviewer #2: At first, I describe my understanding of this paper.

It is important to analyze the structure of social groups in animals such as primates for ecology, and eventually, it may be applied to the human social organization. In this paper, the authors focus on single-file movement, which is often formed by animal groups, and investigate the validity of a method for estimating the structure of groups by reconstructing social networks between individuals based on the adjacency of individuals in single-file movement using multi-agent based simulation. The results show that when the independent agents of a group are somewhat dispersed, it is possible to accurately estimate the cluster structure of the population by observing single-file movement about 30 times.

The points that I feel need to be corrected are noted below. If amendments are made or justifications are shown for not requiring an amendment, I will find this paper acceptable.

- Regarding the overall structure of this paper, there seems to be a replaceability to the content of the introduction and discussion parts. I think there are two candidates for the structure of this paper. The one focuses on single-file movement in general animals as the introduction and discusses the results of the simulation with a focus on primates. Another focuses on the single-file movement in primates as an introduction, discusses the simulation results, and then extends the discussion to the other animals. The current structure is a mixture of both, which makes the position of the model unclear.

- The introduction part seemed to make it difficult to understand the motivation and aims of this study due to the order of explanation. It will be easier to understand if you explain the significance of cluster estimation in animal societies and single-file movement as its clue, and then explain the usefulness of the SNA approach for analyzing such social organization. If you take this order, however, the importance of the topic of single-file movement should be emphasized, and ecological implications should be interpreted for each parameter of the simulation.

- I would like to know what other methods there are for identifying social groups of primates than the observation of single-file movement, and what advantages the proposed method has over them.

- For the method part, this model relies solely on the distance between agents to create a single-file movement. It is preferable to have ecological evidence about this design. The response threshold model that set differences to individual reactivity is often used to explain the formation of group behavior in animals (Bonabeau et al., 1996, https://doi.org/10.1098/rspb.1996.0229 ). If individual agents had parameters such as response thresholds, the predictive power of this model would be expected to be reduced.

6. PLOS authors have the option to publish the peer review history of their article (what does this mean?). If published, this will include your full peer review and any attached files.

Reviewer #1: No

Reviewer #2: **Yes: **Genta Toya

---

## [Author Response · Author response to Decision Letter 0]

26 Oct 2020

Replies to Reviewer #1(Blind)

Thank you very much for your concise comments, which completely sounds positive for our jobs.

● Very interesting work. The paper is well written, and all the four steps of your systems are clear.

Thanks for your evaluation.

● Counting the number of clusters and compare it with the the ground truth seems too naive, however, I do think this a good starting point, given the experimental nature of this work.

Yes, this is just a starting point of this study series which would be extended to the other empirical works.

● A stronger experimental section would help. For example, it is always useful to see the scalability of your system.

Yes, we agree. Now we improve the initial system and will enhance the scalability, e.g., simulation case of the different numbers of the dependent agents for each of independent agents, or simulation under the condition where some agents are masked of agent identity. In fact, we will experience the case where we could not recognize the identities of some animals of the single-file movement in the field. The simulation with the ID-masked condition will be a good simulation for the case of the lack of IDs.

Now we will start to prepare the improved version implementation of these masked-condition simulations, but we will keep the release of the new version of the code in the next study, which will include the evaluation steps using the actual data of the primate single-file movements. We acknowledge the necessity of the further scalability in the discussion section shortly (see LL 357 - 385.).

---

Replies to reviewer #2 (Dr. Genta Toya)

Thank you very much for reading our paper, even if you may be not keen on primate or behavioral ecology. We greatly appreciate that you carefully comment on our study.

● At first, I describe my understanding of this paper.

● It is important to analyze the structure of social groups in animals such as primates

for ecology, and eventually, it may be applied to the human social organization. In this paper, the authors focus on single-file movement, which is often formed by animal groups, and investigate the validity of a method for estimating the structure of groups by reconstructing social networks between individuals based on the adjacency of individuals in single-file movement using multi-agent based simulation. The results show that when the independent agents of a group are somewhat dispersed, it is possible to accurately estimate the cluster structure of the population by observing single-file movement about 30 times.

● The points that I feel need to be corrected are noted below. If amendments are made or justifications are shown for not requiring an amendment, I will find this paper acceptable.

Thanks for your evaluation on our attempts.

● Regarding the overall structure of this paper, there seems to be a replaceability to the content of the introduction and discussion parts. I think there are two candidates for the structure of this paper. The one focuses on single-file movement in general animals as the introduction and discusses the results of the simulation with a focus on primates. Another focuses on the single-file movement in primates as an introduction, discusses the simulation results, and then extends the discussion to the other animals. The current structure is a mixture of both, which makes the position of the model unclear.

OK, we improved to change the structure, your suggesting option 1, e.g., starting from a single-file movement in general animal society, and discussing the possible applications to the primate society as a plausible candidate of the models (see introduction).

● The introduction part seemed to make it difficult to understand the motivation and aims of this study due to the order of explanation. It will be easier to understand if you explain the significance of cluster estimation in animal societies and single-file movement as its clue, and then explain the usefulness of the SNA approach for analyzing such social organization. If you take this order, however, the importance of the topic of single-file movement should be emphasized, and ecological implications should be interpreted for each parameter of the simulation.

We guess it was so difficult for you to find our main motivation in the initial draft due to insufficient explanations particularly in “animal’s single-file movements”. The core of the motivations in this paper is 1) to infer the social (latent) organization in the data-driven way, 2) to propose the additional possibilities of the observable data which have not been tested so far, and 3) to provide the methodological validation for the proposed approaches. The aim 1 is quite important particularly for the wild animal researchers like us, because we often experience the difficulty to obtain the high-quality and massive data from the wild animals, and to perform the experimental operations to test the fundamental parameters underlying the social organization. Accordingly, we always search for the promising data types, together with the technological updates. We proposed a single-file movement for this purpose, namely aim 2. The simulation run here is just to check the methodological validation as the aim 3. This (our demonstration) is a necessary step since nobody empirically tests if the single-file movement data provide sufficient information to infer the social organization. Thus we believe that this is a basic and necessary demonstration.

We first considered the revision following your proposed structure of the introduction section, but it was a bit difficult due to the “tight coupling” between the paragraphs in the initial draft. Thus, we mentioned these objectives described above in the introduction section as an independent paragraph (see LL 70 - 85). We believe that this improves much, and satisfies you.

● I would like to know what other methods there are for identifying social groups of primates than the observation of single-file movement, and what advantages the proposed method has over them.

We raised some examples and its difficulties/problems (see LL 76 - 81 and LL 124 - 126).

● For the method part, this model relies solely on the distance between agents to create a single-file movement. It is preferable to have ecological evidence about this design. The response threshold model that set differences to individual reactivity is often used to explain the formation of group behavior in animals (Bonabeau et al., 1996, https://doi.org/10.1098/rspb.1996.0229). If individual agents had parameters such as response thresholds, the predictive power of this model would be expected to be reduced.

Thanks. This is an important point, and limitation of the current demonstrations. As you raised, the “real” animal would not make a decision simply depending on the inter-individual spatial distance. They may make a movement decision in a non-linear way, e.g., the distance threshold nonlinearly activates the movement decision, as you suggested above in the previous model approach. The present model only tests the limited condition of the agent's spatial distribution. Now we will prepare to extend the another situation where the probabilistic threshold is considered for movement-decision making, or ID-masked conditions fitted with the realistic field data. We clearly mentioned the present research limitations of our demonstration and further acknowledge the possible attempts on these proposed parameter conditions for the generalization of the animal agent model to the realistic wildlife observation (see LL 358 - 368).

---

## [Decision Letter · Decision Letter 1]

17 Nov 2020

Agent-based simulation for reconstructing social structure by observing collective movements with special reference to single-file movement

PONE-D-20-20675R1

Dear Dr. Koda,

We’re pleased to inform you that your manuscript has been judged scientifically suitable for publication and will be formally accepted for publication once it meets all outstanding technical requirements.

Kind regards,

Ilya Safro, Ph.D.

Academic Editor

PLOS ONE

Additional Editor Comments (optional):

Reviewers' comments:

Reviewer's Responses to Questions

**Comments to the Author**

1. If the authors have adequately addressed your comments raised in a previous round of review and you feel that this manuscript is now acceptable for publication, you may indicate that here to bypass the “Comments to the Author” section, enter your conflict of interest statement in the “Confidential to Editor” section, and submit your "Accept" recommendation.

Reviewer #2: All comments have been addressed

2. Is the manuscript technically sound, and do the data support the conclusions?

Reviewer #2: Yes

3. Has the statistical analysis been performed appropriately and rigorously? 

Reviewer #2: Yes

4. Have the authors made all data underlying the findings in their manuscript fully available?

Reviewer #2: (No Response)

5. Is the manuscript presented in an intelligible fashion and written in standard English?

Reviewer #2: Yes

6. Review Comments to the Author

Reviewer #2: (No Response)

7. PLOS authors have the option to publish the peer review history of their article (what does this mean?). If published, this will include your full peer review and any attached files.

Reviewer #2: **Yes: **Genta Toya

---

## [Editor Report · Acceptance letter]

23 Nov 2020

PONE-D-20-20675R1 

Agent-based simulation for reconstructing social structure by observing collective movements with special reference to single-file movement 

Dear Dr. Koda:

I'm pleased to inform you that your manuscript has been deemed suitable for publication in PLOS ONE. Congratulations! Your manuscript is now with our production department. 

Kind regards, 

on behalf of

Dr. Ilya Safro 

Academic Editor

PLOS ONE